# IMPLICIT GENERATIVE MODELING FOR EFFICIENT EXPLORATION

## ABSTRACT

Efficient exploration remains a challenging problem in reinforcement learning, especially for those tasks where rewards from environments are sparse. A commonly used approach for exploring such environments is to introduce some "intrinsic" reward. In this work, we focus on model uncertainty estimation as an intrinsic reward for efficient exploration. In particular, we introduce an implicit generative modeling approach to estimate a Bayesian uncertainty of the agent's belief of the environment dynamics. Each random draw from our generative model is a neural network that instantiates the dynamic function, hence multiple draws would approximate the posterior, and the variance in the future prediction based on this posterior is used as an intrinsic reward for exploration. We design a training algorithm for our generative model based on the amortized Stein Variational Gradient Descent. In experiments, we compare our implementation with state-of-the-art intrinsic reward-based exploration approaches, including two recent approaches based on an ensemble of dynamic models. In challenging exploration tasks, our implicit generative model consistently outperforms competing approaches regarding data efficiency in exploration.

## 1 INTRODUCTION

Reinforcement learning (RL) has enjoyed recent success in a variety of applications, including super-human performance in Atari games (Mnih et al., 2013), robotic control (Lillicrap et al., 2015), image-based control tasks (Hafner et al., 2019), and playing the game of Go (Silver et al., 2016). Despite these achievements, many recent RL techniques still suffer from poor sample efficiency. Agents are often trained for millions, or even billions of simulation steps before achieving a reasonable performance (Burda et al., 2018a). This lack of statistical efficiency makes it difficult to apply RL to real-world tasks, as the cost of acting in the real world is far greater than in a simulator. It is then a problem of utmost importance, to design agents that make efficient use of collected data. According to (Sutton & Barto, 2018), there are three key aspects in building a data-efficient agent for reinforcement learning: generalization, exploration, and long-term consequence awareness. In this work, we focus on the efficient exploration aspect. In particular, we focus on those challenging environments with sparse external rewards, where exploration is commonly driven by some sort of intrinsic reward. It is observed (Osband et al., 2017; 2018) that a Bayesian uncertainty[1] estimate plays an important role in efficient exploration in deep RL, but is unfortunately not appropriately addressed in the majority of state-of-the-art RL algorithms.

In this work, we introduce a new framework of Bayesian uncertainty modeling for intrinsic reward-based exploration in RL. Our framework characterizes the (epistemic) uncertainty in the agent's belief of the environment dynamics in a non-parametric way to enable flexibility and expressiveness. The main component of our framework is a network generator, each draw of which is a neural network that serves as the dynamic function for RL. Multiple draws then approximate a posterior of the dynamic model and the variance in future state prediction based on this posterior is used as an intrinsic reward for exploration. Recently, it has been shown (Ratzlaff & Fuxin, 2019) that training such kind of generators can be done in classification problems and the resulting draws of

---

[1]As noted in (Osband et al., 2018), it is the (epistemic) uncertainty in agent's belief, rather than the (aleatoric) uncertainty of outcome which reflects the inherent randomness of the environment, that matters the most regarding efficient exploration in RL.

networks can represent a rich distribution of networks that perform approximately equally well on the classification task. For our goal of training this generator for the dynamic function, we propose an algorithm to optimize the KL divergence between the implicit distribution (represented by draws from the generator) and the true posterior of the dynamic model (given the agent's experience) via the amortized Stein Variational Gradient Descent (SVGD) (Liu & Wang, 2016; Feng et al., 2017).

Comparing with recent works (Pathak et al., 2019; Shyam et al., 2019) that maintain an ensemble of dynamic models and use the divergence or disagreement among them as an intrinsic reward for exploration, our implicit modeling of the posterior has several advantages: Firstly, it is a more flexible framework for approximating the model posterior comparing with ensemble-based approximation where the number of particles is fixed. Secondly, it is based on the principle of amortized SVGD (Feng et al., 2017), where the KL divergence between the implicit posterior and the true posterior is directly minimized in a nonparametric sense, and further projected to a finite-dimensional parameter update. This is in contrast with existing ensemble-based methods that count on the random initialization and/or bootstrapped experience sampling for the ensemble to approximate the posterior. Thirdly, it is more memory efficient given that our method stores and updates only parameters of the generator, in contrast with parameters of every member network of the ensemble.

In our experiments, we compare our approach with several state-of-the-art intrinsic reward-based exploration approaches, including two recent approaches that also leverage the uncertainty in dynamic models. In all the tasks we have tested, our implementation consistently outperforms competing methods regarding data efficiency in exploration.

In summary, our contributions are:

- We propose a new framework for implicitly approximating the posterior of network parameters where the uncertainty of the network function can be used as an intrinsic reward for efficient exploration in RL.

- We design an amortized SVGD-based training algorithm for the proposed framework and apply it to approximate the implicit distribution of the dynamic model of the environment.

- We test our implementation on three challenging exploration tasks and compare with three state-of-the-art intrinsic reward-based methods, two of which are also based on uncertainty in dynamic models. The consistent superior performance of our method demonstrates the effectiveness of the proposed framework in estimating Bayesian uncertainty in the dynamic model for efficient exploration.

## 2 PROBLEM SETUP AND BACKGROUND

Consider a Markov Decision Process (MDP) represented as $(\mathcal{S}, \mathcal{A}, P, r, \rho_0)$, where $\mathcal{S}$ is the state space, $\mathcal{A}$ is the action space. $P : \mathcal{S} \times \mathcal{A} \times \mathcal{S} \to [0, 1]$ is the unknown dynamics model, specifying the probability of transitioning to next state $s'$ from current state $s$ by taking action $a$, as $P(s'|s, a)$. $r : \mathcal{S} \times \mathcal{A} \to \mathbb{R}$ is the reward function, $\rho_0 : \mathcal{S} \to [0, 1]$ is the distribution of initial states. A policy is a function $\pi : \mathcal{S} \times \mathcal{A} \to [0, 1]$, which outputs a distribution over the action space for given state $s$.

### 2.1 EXPLORATION IN REINFORCEMENT LEARNING

In online decision-making problems, such as multi-arm bandits and reinforcement learning, a fundamental dilemma in an agent's choice is exploitation versus exploration. Exploitation refers to making the best decision given current information, while exploration refers to gathering more information about the environment. In standard reinforcement learning setting where the agent receives an external reward for each transition step, common recipes for exploration/exploitation trade-off include naive methods such as $\epsilon$-greedy (Sutton & Barto, 2018) and optimistic initialization (Lai & Robbins, 1985), posterior guided methods such as upper confidence bounds (Auer, 2002; Dani et al., 2008) and Thompson sampling (Thompson, 1933). In the situation we focus on, where external rewards are sparse or disregarded, the above trade-off narrows down to the pure exploration problem of efficiently accumulating information about the environment. The common approach is to explore in a task-agnostic way under some "intrinsic" reward. An exploration policy can then be trained in the standard RL way where dense rewards are available. Existing methods construct intrinsic rewards from visitation frequency of the state (Bellemare et al., 2016), prediction error of the dynamic model as "curiosity" (Pathak et al., 2017), diversity of visited states (Eysenbach et al., 2018), etc.

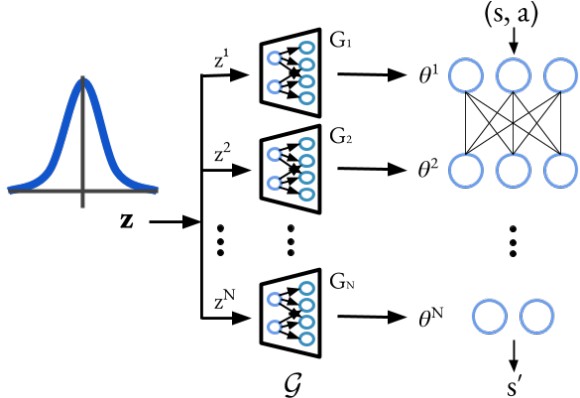

Figure 1: Architecture of the layer-wise generator of the dynamic model. Independent noise samples $\{z^1, \cdots, z^N\}$ are drawn from a standard Gaussian with diagonal covariance, and input to layer-wise generators $\{G_1, \cdots, G_N\}$. Each generator $G_j$ outputs parameters $\theta^j$ for the corresponding $j$-th layer of the neural network representing the dynamic model.

## 2.2 DYNAMIC MODEL UNCERTAINTY AS INTRINSIC REWARD

Following the guiding principle of modeling Bayesian uncertainty in online decision making, two recent methods (Pathak et al., 2019; Shyam et al., 2019) train an ensemble of dynamic models and use the variation/information gain as an intrinsic reward for exploration. In this work, we follow the similar idea of exploiting the uncertainty in the the dynamic model, but emphasize on the implicit posterior modeling in contrast with directly training an ensemble of dynamic models.

Let $f : \mathcal{S} \times \mathcal{A} \to \mathcal{S}$ denote a model of the environment dynamics (usually represented by a neural network) we want to learn based on the agent experience $\mathcal{D}$. We design a generator module $\mathcal{G}$ which takes a random draw from the normal distribution and outputs a sample vector of parameters $\boldsymbol{\theta}$ that determines $f$ (denoted as $f_{\boldsymbol{\theta}}$). If samples from $\mathcal{G}$ represent the posterior distribution $p(f_{\boldsymbol{\theta}}|D)$, then given $(s_t, a_t)$, the uncertainty in the output of the dynamics model can be computed by the following variance among a set of samples $\{\boldsymbol{\theta}_i\}_{i=1}^m$ from $\mathcal{G}$, and used as an intrinsic reward $r^{in}$ for learning an exploration policy,

$$r_t^{in} = \sum\nolimits_{i=1}^m \left\| f_{\boldsymbol{\theta}_i}(s_t, a_t) - \frac{1}{m} \sum\nolimits_{\ell=1}^m f_{\boldsymbol{\theta}_\ell}(s_t, a_t) \right\|^2. \tag{1}$$

In learning the exploration policy, this intrinsic reward can be computed with either actual rollouts in the environment or simulated rollouts generated by the estimated dynamic model.

## 3 POSTERIOR APPROXIMATION VIA AMORTIZED SVGD

In this section, we introduce the core component of our exploration agent, the dynamic model generator $\mathcal{G}$. In the following subsections, we first introduce the design of this generator and then describe its training algorithm in detail. A summary of our algorithm is given in the last subsection.

### 3.1 IMPLICIT POSTERIOR GENERATOR

As shown in Fig. 1, the dynamic model is defined as a $N$-layer neural network function $f_{\boldsymbol{\theta}}(s, a)$, with input (state, action) pair $(s, a)$ and model parameters $\boldsymbol{\theta} = (\theta^1, \cdots, \theta^N)$, where $\theta^j$ represents network parameters of the $j$-th layer. The generator module $\mathcal{G}$ consists of exactly $N$ layer-wise generators, $\{G_1, \cdots, G_N\}$, where each $G_j$ takes a random noise vector $z^j \in \mathbb{R}^d$ and outputs the corresponding parameter vector $\theta^j = G_j(z^j; \eta^j)$, where $\eta^j$ are the parameters of $G_j$. Note that $z^j$'s are generated independently from a $d$-dimensional standard normal distribution, rather than jointly.

As mentioned in §1, this framework has advantages in flexibility and efficiency, comparing with ensemble-based methods (Shyam et al., 2019; Pathak et al., 2019), since it maintains only parameters of the $N$ generators, i.e., $\boldsymbol{\eta} = (\eta^1, \cdots, \eta^N)$, and enables drawing an arbitrary number of sample networks to approximate the posterior of the dynamic model.

### 3.2 TRAINING WITH AMORTIZED STEIN VARIATIONAL GRADIENT DESCENT

We now introduce the training algorithm of the generator module $\mathcal{G}$. Assuming that the true posterior of the dynamic model given agent's experience $\mathcal{D}$ is $p(f|\mathcal{D})$, and the implicit distribution of $f_{\boldsymbol{\theta}}$

captured by $\mathcal{G}$ is $q(f_{\boldsymbol{\theta}})$. We want $q(f_{\boldsymbol{\theta}})$ to be as close as possible to $p(f|\mathcal{D})$, such closeness is commonly measured by the KL divergence $\mathbf{D}_{\mathrm{KL}}[q(f_{\boldsymbol{\theta}})\|p(f|\mathcal{D})]$. Traditional approach for finding $q$ that minimizes $\mathbf{D}_{\mathrm{KL}}[q(f_{\boldsymbol{\theta}})\|p(f|\mathcal{D})]$ is variational inference, where an evidence lower bound (ELBO) is maximized. Recently, a nonparametric variational inference framework, Stein Variational Gradient Descent (SVGD) (Liu & Wang, 2016), was proposed, which represents $q$ with a set of particles rather than making any parametric assumptions, and approximates the functional gradient descent w.r.t. $\mathbf{D}_{\mathrm{KL}}[q(f_{\boldsymbol{\theta}})\|p(f|\mathcal{D})]$ by iterative particle evolvement. We apply SVGD to our sampled network functions, and follow the idea of amortized SVGD (Feng et al., 2017) to project the functional gradients to the parameter space of $\boldsymbol{\eta}$ by back-propagation through the generators.

Given a set of dynamic functions $\{f_{\boldsymbol{\theta}_i}\}_{i=1}^{m}$ sampled from $\mathcal{G}$, SVGD updates each function by

$$f_{\boldsymbol{\theta}_i} \leftarrow f_{\boldsymbol{\theta}_i} + \epsilon\phi^*(f_{\boldsymbol{\theta}_i}), \qquad i = 1, \cdots, m,$$

where $\epsilon$ is a step size, and $\phi^*$ is the function in the unit ball of a reproducing kernel Hilbert space (RKHS) $\mathcal{H}$ that maximally decreases the KL divergence between the distribution $q$ represented by $\{f_{\boldsymbol{\theta}_i}\}_{i=1}^{m}$ and the target posterior $p$,

$$\phi^* = \max_{\phi \in \mathcal{H}} \left\{ -\frac{d}{d\epsilon}\mathbf{D}_{\mathrm{KL}}(q\|p), \quad s.t. \|\phi\|_{\mathcal{H}} \leq 1 \right\}.$$

This optimization problem has a closed form solution,

$$\phi^*(f_{\boldsymbol{\theta}_i}) = \mathbb{E}_{f \sim q}\left[\nabla_f \log p(f) k(f, f_{\boldsymbol{\theta}_i}) + \nabla_f k(f, f_{\boldsymbol{\theta}_i})\right],$$

where $k(\cdot, \cdot)$ is the positive definite kernel associated with the RKHS. We use a Gaussian kernel for our implementation. The log-likelihood term for $f_{\boldsymbol{\theta}}$ corresponds to the negation of the regression loss of future state prediction for all transitions in $\mathcal{D}$, i.e., $-\mathcal{L}(f_{\boldsymbol{\theta}}) = -\sum_{(s,a,s') \in \mathcal{D}} L(f_{\boldsymbol{\theta}}(s, a), s')$. Given that $f_{\boldsymbol{\theta}}$ is determined by $\boldsymbol{\theta}$, the corresponding SVGD update rule for each sampled $\boldsymbol{\theta}_i$ is,

$$\boldsymbol{\theta}_i \leftarrow \boldsymbol{\theta}_i + \epsilon\phi^*(\boldsymbol{\theta}_i), \qquad i = 1, \cdots, m,$$

where

$$\phi^*(\boldsymbol{\theta}_i) = \mathbb{E}_{\boldsymbol{\theta} \sim \mathcal{G}}\left[-\nabla_{\boldsymbol{\theta}}\mathcal{L}(f_{\boldsymbol{\theta}}) k(\boldsymbol{\theta}, \boldsymbol{\theta}_i) + \nabla_{\boldsymbol{\theta}} k(\boldsymbol{\theta}, \boldsymbol{\theta}_i)\right]. \tag{2}$$

Given that $\boldsymbol{\theta}_i$'s are generated by $\mathcal{G}(\boldsymbol{z}; \boldsymbol{\eta})$, the update rule for $\boldsymbol{\eta}$ can be obtained by by the chain rule,

$$\boldsymbol{\eta} \leftarrow \boldsymbol{\eta} + \epsilon\sum_{i=1}^{m} \nabla_{\boldsymbol{\eta}}\mathcal{G}(\boldsymbol{z}_i; \boldsymbol{\eta})\phi^*(\boldsymbol{\theta}_i), \tag{3}$$

where $\phi^*(\boldsymbol{\theta}_i)$ can be computed by (2) using empirical expectation from sampled batch $\{\boldsymbol{\theta}_i\}_{i=1}^{m}$,

$$\phi^*(\boldsymbol{\theta}_i) = \frac{1}{m}\sum_{\ell=1}^{m}\left\{-\left[\sum_{(s,a,s') \in \mathcal{D}} \nabla_{\boldsymbol{\theta}_\ell} L(f_{\boldsymbol{\theta}_\ell}(s, a), s')\right] k(\boldsymbol{\theta}_\ell, \boldsymbol{\theta}_i) + \nabla_{\boldsymbol{\theta}_\ell} k(\boldsymbol{\theta}_\ell, \boldsymbol{\theta}_i)\right\}. \tag{4}$$

---

**Algorithm 1:** Exploration with an Implicit Distribution

**Initialize** Generator $\mathcal{G}_{\boldsymbol{\eta}}$, parameters $T, m$
**Initialize** Policy $\pi$, Experience buffer $\mathcal{D}$
**while** True **do**
    **while** episode not **done: do**
        $f_{\boldsymbol{\Theta}} \leftarrow \mathcal{G}(\boldsymbol{z}; \boldsymbol{\eta}), \boldsymbol{z} \sim \mathcal{N}(0, I^d)$
        $\boldsymbol{\eta} \leftarrow$ evaluate (3), (4) on $\mathcal{D}$
        $\mathcal{D}_{\pi} \leftarrow \mathcal{D}$ or $\widetilde{\mathcal{D}} \sim \mathrm{MDP}(f_{\boldsymbol{\Theta}})$
        $R_{\pi} \leftarrow r^{in}(f_{\boldsymbol{\theta}}, s, a | (s, a) \sim \mathcal{D}_{\pi})$ by (1)
        $\pi \leftarrow$ update policy on $(\mathcal{D}_{\pi}, R_{\pi})$
        $\mathcal{D}_T \leftarrow$ rollout $\pi$ for $T$ steps
        $\mathcal{D} \leftarrow \mathcal{D} \cup \mathcal{D}_T$
    **end**
**end**

---

Algorithm 1 shows our procedure in psuedocode. Starting with a buffer $\mathcal{D}$ of random transitions, our algorithm samples a set of dynamic models $f_{\boldsymbol{\Theta}} = \{f_{\boldsymbol{\theta}_i}\}$ from the generator $\mathcal{G}$, and updates the generator parameters $\boldsymbol{\eta}$ using amortized SVGD (3) and (4). For policy update, the intrinsic reward (1) is evaluated on either the actual experience $\mathcal{D}$ or the simulated experience $\widetilde{\mathcal{D}}$ generated by $f_{\boldsymbol{\theta}_i}$. The exploration policy is then updated using a model-free RL algorithm on the collected experience $\mathcal{D}_{\pi}$ and intrinsic rewards $R_{\pi}$. The updated exploration policy is then used to rollout in the environment for $T$ steps so that new transitions are collected and added to the buffer $\mathcal{D}$ for subsequent iterations.

### 3.3 SUMMARY OF THE EXPLORATION ALGORITHM

To condense what we have proposed so far, we summarize in Algorithm 1 the procedure used to train the generator of dynamic models and the exploration policies. We repeat the process, with the agent acting in the environment under the exploration policy and collecting new experience.

## 4 EXPERIMENTS

In this section we conduct experiments to compare our approach to existing state-of-the-art in efficient exploration with intrinsic reward. For our propose, only the task-agnostic setting is considered, where the agent explores the environment irrespective of the downstream task. Task agnostic exploration is essential when external rewards are sparse and there is large uncertainty in the environment.

### 4.1 TOY TASK: NCHAIN

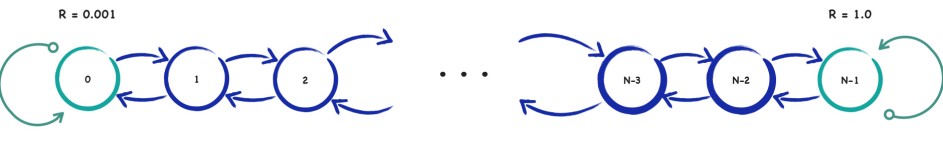

Figure 2: NChain task.

As a sanity check, we first follow *MAX* (Shyam et al., 2019) to evaluate our method on a stochastic version of the toy environment NChain. As shown in Figure 2, the chain is a finite sequence of $N$ states. Each episode starts from state 1 and lasts for $N + 9$ steps. For each step, the agent can move forward to the next state in the chain or backward to the previous state. Attempting to move off the edge of the chain results in the agent staying still. Reward is only afforded to the agent at the edge states: $0.01$ for reaching state $0$, and $1.0$ for reaching state $N - 1$. In addition, there is uncertainty built into the environment: each state is designated as a *flip-state* with probability $0.5$. When acting from a flip-state, the agent's actions are reversed, i.e., moving forward will result in movement backward, and vice-versa. Given the (initially) random dynamics and a sufficiently long chain, we expect an agent using an $\epsilon$-greedy exploration strategy to exploit only the small reward of state $0$. In contrast, agents with exploration policies which actively reduce uncertainty can efficiently discover every state in the chain. Figure 7 shows that our agent navigates the chain in less than 15 episodes, while indeed, the $\epsilon$-greedy agent (double DQN) does not make meaningful progress. For completeness, we also evaluate the agents introduced in the following sections, and show the results in the appendix.

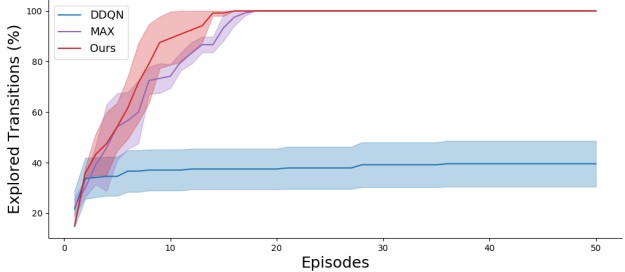

Figure 3: Results on the 40-link chain environment. Each line is the mean of three runs, with the shaded regions corresponding to $\pm 1$ standard deviation. Both our method and *MAX* actively reduce uncertainty in the chain, and therefore are able to quickly explore to the end of the chain. $\epsilon$-greedy DDQN fails to explore more than 40% of the chain.

## 4.2 CONTINUOUS CONTROL ENVIRONMENTS

We also consider three challenging continuous control tasks in which efficient exploration is known to be difficult. In each environment, the dynamics are nonlinear and cannot be solved with simpler (efficient) tabular approaches. As stated above, external reward is completely removed; the agent is motivated purely by the uncertainty in its belief of the environment.

**Experimental setup**

To validate the effectiveness of our method, we compare with several state-of-the-art formulations of intrinsic reward. Specifically, we conduct experiments comparing the following methods:

- (*Ours*) The proposed intrinsic reward, using the estimated variance of an implicit distribution of the dynamic model.
- (*ICM*) Error between predicted next state and observed next state (Pathak et al., 2017).
- (*Disagreement*) Variance of predictions from an ensemble of dynamic models (Pathak et al., 2019).
- (*MAX*) Jensen-Renyi information gain of the dynamic function (Shyam et al., 2019).
- (*Random*) Pure random exploration as a naive baseline.

**Implementation details**

Given our goal is to compare the performance across different intrinsic rewards, we fix the model architecture, training pipeline, and hyperparameters across all methods.[2] The dynamic models are 4 layer fully connected neural networks. For the purpose of computing the information gain, dynamic models for *MAX* predict both mean and variance of the next state, while for other methods, Dynamic models predict only the mean. Our generator as well as the dynamic models for other methods are optimized using Adam (Kingma & Ba, 2014) with a learning rate of $1e^{-4}$. To learn exploration policies, we use the Soft Actor Critic (Haarnoja et al., 2018) algorithm for all methods. For *MAX*, *ICM*, and *Disagreement*, we use ensembles of 32 dynamic models respectively to compute the intrinsic reward. Since our method trains a generator of dynamic models instead, we fix the number of models we sample from the generator at $m = 32$ for a fair comparison. Further implementation details can be found in the supplementary material.

### 4.2.1 ACROBOT CONTROL

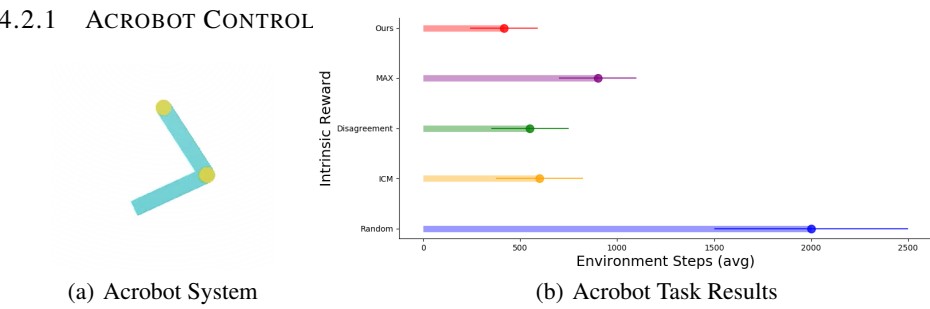

(a) Acrobot System      (b) Acrobot Task Results

Figure 4: (a) The Acrobat system. (b) Performance of each method on the Acrobot environment (average of five seeds), with error bars representing ± one standard deviation. The length of each horizontal bar indicates the number of environment steps each agent/method takes to swing the acrobot to fully horizontal on both (left and right) directions.

Our first environment is a modified continuous control version of the Acrobot. As shown in Figure 4(a), the Acrobot environment begins with a hanging down pendulum which consists of two links connected by an actuated joint. Normally, an action $a \in \{-1, 0, 1\}$ applies or not ($a = 0$) a unit force on the joint in the left or right direction. We modify the environment such that a continuous action $a \in [-1, 1]$ applies a force $F = |a|$ in the corresponding direction.

---

[2]We use the codebase of *MAX* as a basis and implement *Ours*, *ICM*, and *Disagreement* intrinsic rewards under the same framework.

To focus on efficient exploration, we test the ability of each exploration method to sweep the entire lower hemisphere: positioning the acrobot completely horizontal towards both (left and right) directions. Given this is a relatively simple task and can be solved by random exploration, as shown in Figure 4(b), all four intrinsic reward methods solve it within just hundreds of steps and our method is the most efficient one. The takeaway here is that in relatively simple environments where there might be little room for improvement over state-of-the-art, our method still achieves a better performance due to its flexibility and efficiency in approximating the model posterior. We will see in subsequent experiments that this observation scales well with more difficult environments.

### 4.2.2 ANT MAZE NAVIGATION

Next, we evaluate on the Ant Maze environment (Figure 5(a)). In this control task, the agent provides torques to each of the 8 joints of the ant. The provided observation contains the pose of the torso as well as the angles and velocities of each joint. The agent's performance is measured by the percentage of the U-shaped maze explored during evaluation. Figure 5(b) shows the result of each method over 5 seeds. Our agent consistently navigates to the end of the maze at the time when other methods have only explored 60% or less. We show how state visitation frequencies progress through training in figures 5(c)-5(f). While *MAX* (Shyam et al., 2019) also navigates the maze, the more advanced uncertainty modeling of our method allows our agent to better estimate the state novelty, which leads to a considerably quicker exploration.

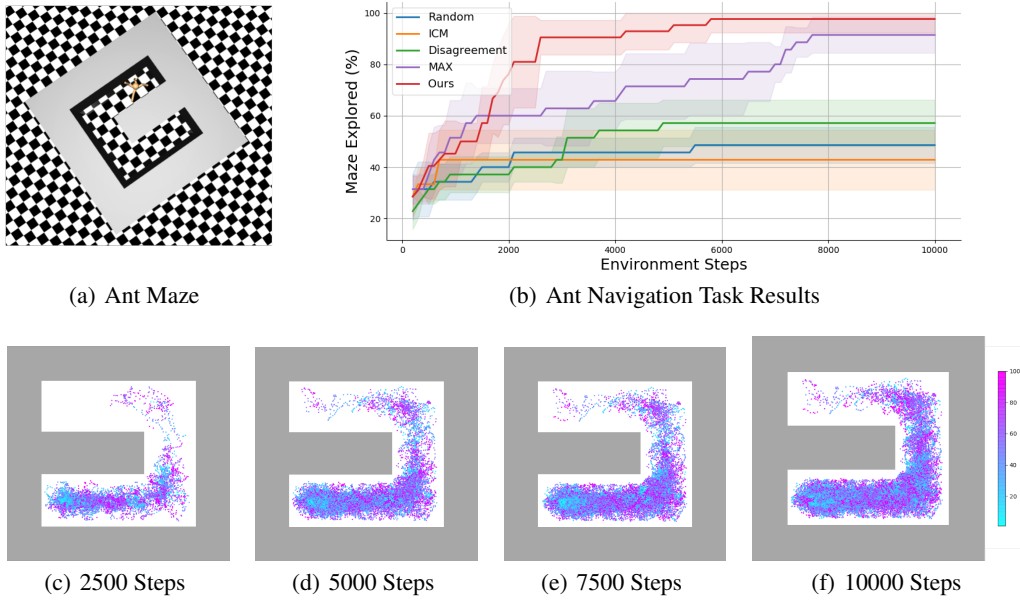

(a) Ant Maze            (b) Ant Navigation Task Results

(c) 2500 Steps     (d) 5000 Steps     (e) 7500 Steps     (f) 10000 Steps

Figure 5: (a) The Ant Maze environment. (b) Performance of each method with mean and $\pm 1$ standard deviation (shaded region) over five seeds. $x$-axis is the number of steps the ant has moved, $y$-axis is the percentage of the U-shaped maze that has been explored. Our method (red) is able to navigate to the end of the maze much faster than any other method. Figures (c-f) show the behavior of the agent at different stages of training. Points are color-coded with blue points occurring at the beginning of the episode, and red points at the end.

### 4.2.3 ROBOTIC MANIPULATION

The final task is an exploration task in a robotic manipulation environment, HandManipulateBlock. As shown in Figure 6(a), a robotic hand is given a palm-sized block for manipulation. The agent has actuation control of the 20 joints that make up the hand, and its exploration performance is measured

by the percentage of possible rotations of the cube that the agent performs.[3] In particular, the state of the cube is represented by Cartesian coordinates along with a quaternion to represent the rotation. We transform the quaternion to Euler angles and discretize the resulting state space by 45 degree intervals. The agent is evaluated based on how many of the 512 total states are visited.

This task is far more challenging than previous tasks, having a larger state space and action space. Additionally, states are more difficult to reach than the Ant Maze environment; requiring manipulation of 20 joints instead of 8. In order to explore in this environment, an agent must also learn how to rotate the block without dropping it. Figure 6(b) shows the performance of each method over 5 seeds. This environment proved very challenging for all methods, none succeeded in exploring more than half of the state space. When placed in a complicated environment where the task is not clear, we want our agents to explore as fast as possible, in order to master the dynamics of the environment. For this environment, we can see that our method indeed performs the best by a clear margin, regarding exploration efficiency.

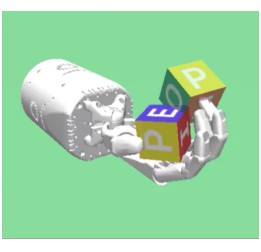

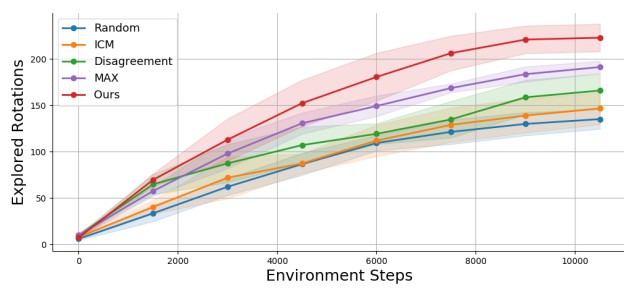

(a) Robotic Hand          (b) Manipulation Task Results

Figure 6: (a) The Robotic Hand task in motion. (b) Performance of each method with mean and $\pm 1$ standard deviation (shaded region) over five seeds. $x$-axis is the number of manipulation steps, $y$-axis is the number of rotation states of the block that has been explored. Our method (red) explores clearly faster than all other methods.

## 5 RELATED WORK

Efficient Exploration remains a major challenge in deep reinforcement learning (Fortunato et al., 2017; Burda et al., 2018b; Eysenbach et al., 2018; Burda et al., 2018a), and there is no consensus on the *correct* way to explore an environment. One practical guiding principle for efficient exploration is the reduction of agent's epistemic uncertainty of the environment (Chaloner & Verdinelli, 1995; Osband et al., 2017). Osband et al. (2016) uses a bootstrap ensemble of DQNs, where the predictions of the ensemble are used as an estimate of the agent's uncertainty over the value function. Osband et al. (2018) proposed to augment the predictions of a DQN agent by adding the contribution from a prior to the value estimate. In contrast to our method, these approaches seek to estimate the uncertainty in the *value function*, while we focus on exploration with intrinsic reward by estimating the uncertainty of the *dynamic model*. Fortunato et al. (2017) add parameterized noise to the agent's weights, to induce state-dependant exploration beyond $\epsilon$-greedy or entropy bonus.

Methods for constructing intrinsic rewards for exploration have become the subject of increased study. One well-known approach is to use the prediction error of an inverse dynamics model as an intrinsic reward (Pathak et al., 2017; Schmidhuber, 1991). Schmidhuber (1991) and Sun et al. (2011) proposed using the learning progress of the agent as an intrinsic reward. Count based methods (Bellemare et al., 2016; Ostrovski et al., 2017) give a reward proportional to the visitation count of a state. Houthooft et al. (2016) formulate exploration as a variational inference problem, and use Bayesian neural networks (BNN) to maintain the agent's belief over the transition dynamics. The BNN predictions are used to estimate a form of Bayesian information gain called compression improvement. The variational approach is also explored in Mohamed & Rezende (2015); Gregor et al.

---

[3]This is different from the original goal of this environment since we want to evaluate task-agnostic exploration rather than goal-based policies.

(2016); Salge et al. (2014), who proposed using intrinsic rewards based on a variational lower bound on empowerment; the mutual information between an action and the induced next state. This reward is used to learn a set of discriminative low-level skills. The most closely-related work to ours are two recent methods (Pathak et al., 2019; Shyam et al., 2019) that compute intrinsic rewards from an ensemble of dynamic models. Disagreement among the ensemble members in next-state predictions is computed as an intrinsic reward. Shyam et al. (2019) also uses *active exploration* (Schmidhuber, 2003; Chua et al., 2018), in which the agent is trained in a surrogate MDP, to maximize intrinsic reward before acting in the real environment. Our method follows the similar idea of exploiting the uncertainty in the dynamic model, but instead suggests an implicit generative modeling of the posterior of the dynamic function, which enables a more flexible approximation of the posterior uncertainty with better sample efficiency.

There has been a wealth of research on nonparametric particle-based variational inference methods (Liu & Wang, 2016; Dai et al., 2016; Ambrogioni et al., 2018), where a certain number of particles are maintained to represent the variational distribution, and updated by solving an optimization problem. Notably, we make use of the amortized SVGD (Feng et al., 2017) to optimize our generator for approximately sampling from the posterior of the dynamic model.

## 6 CONCLUSION

In this work, we introduced a novel method for representing the agent's uncertainty of the environment dynamics. We formulated an intrinsic reward based on the uncertainty given by an approximate posterior of the dynamic model to enable efficient exploration in difficult environments, Through experiments in control, navigation, and manipulation, we demonstrated that our method is consistently more sample efficient than the baseline methods. Future work includes investigating the efficacy of learning an approximate posterior of the agent's value or policy model, as well as more efficient sampling techniques.

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

## A  APPENDIX

### A.1  IMPLEMENTATION DETAILS FOR CONTINUOUS ENVIRONMENTS

Here we describe in more detail the various implementation choices we used for our method as well as for the baselines.

**Toy Chain Environment**

The chain environment was implemented based on the NChain-v0 gym environment. We altered NChain-v0 to contain 40 states instead of 10 to reduce the possibility of solving the environment with random actions. We also modified the stochastic 'slipping' state behavior by fixing the behavior of the states respect to reversing an action. For both our method and *MAX*, we use ensembles of 5 deterministic neural networks with 4 layers, each are 256 units wide with tanh nonlinearities. As usual, our ensembles are sampled from the generator at each timestep, while *MAX* uses a static ensemble. We generate each layer in the target network with generators composed of two hidden layers, 64 units each with ReLU nonlinearities. Both models are trained by minimizing the regression loss on the observed data. We optimize using Adam with a learning rate of $10^{-4}$, and weight decay of $10^{-6}$. We use Monte Carlo Tree Search (MCTS) to find exploration policies for use in the environment. We build the tree with 25 iterations of 10 random trajectories, and UCB-1 as the selection criteria. Crucially, during rollouts, we query the dynamic models instead of the simulator, and we compute the corresponding intrinsic reward. For *MAX* we use the Jensen Shannon divergence while our method uses the variance in the predictions of our samples. There is a small discrepancy between the numbers reported in the MAX paper for the chain environment. This is due to using UCB-1 as the selection criteria instead of Thompson sampling as used in the MAX paper. We take actions in the environment based on the children with the highest value. The tree is then discarded after one step, after which, the dynamic models are fit for 10 additional epochs.

**Continuous Control Environments**

For each method, we kept the implementation details consistent across the environments Acrobot, Ant Maze, and Block Manipulation. The common details of each exploration method are as follows.

Each method uses (or samples) ensembles to approximate environment dynamics. Models in the ensemble are composed of 32 networks with 4 hidden layers, 512 units wide with ReLU nonlinearities, except for *MAX* which uses swish[4]. *ICM*, *Disagreement*, and our method use ensembles of deterministic models, while *MAX* uses probabilistic networks which output a Gaussian distribution over next states. The approximate dynamic models (ensembles/generators) are optimized with Adam, using a minibatch size of 256, a learning rate of $10^{-4}$, and weight decay of $10^{-4}$.

For our dynamic model, each layer generator is composed of two hidden layers, 64 units wide and ReLU nonlinearity. The output dimensionality of each generator is equal to the product of the input and output dimensionality of the corresponding layer in the dynamic model. To sample one dynamic model, each generator takes as input an independent draw from $z \sim \mathcal{Z}$ where $\mathcal{Z} = \mathcal{N}(0, I^{32})$ where $I$ is a 32 dimensional identity matrix. We sample ensembles of arbitrary size $m$ by instead providing a batch $\{z\}_{i=1}^{m}$ as input. To train the generator such that we can sample accurate transition models we update according to Equation 4; we compute the regression error on the data, as well as the repulsive term using an appropriate kernel. For all experiments we used a standard Gaussian kernel $k(\theta, \theta_i) = \exp\left(-\frac{1}{h}||\theta - \theta_i||_2^2\right)$ where $h$ is the median of the pairwise distances between sampled particles $\{\theta\}_{i=1}^{m}$. Because we sample functions $f_{\theta}$ instead of data points, the pairwise distance is computed by using the likelihood of the data $x$ under the model: $\log f_{\theta}(x)$.

For *MAX*, we used the code provided from (Shyam et al., 2019). Each member in the approximate dynamic model ensemble is a probabilistic neural network that predicts a Gaussian distribution (with diagonal covariance) over next states. The exploration policies are trained with SAC, given an experience buffer of rollouts $\bar{D} = \{s, a, s'\} \cup R\pi$ performed by dynamic models, where $R_{\pi}$ is the intrinsic reward: the Jensen-Renyi divergence between next state predictions of the dynamic model. the policy trained with SAC acts in the environment to maximize the intrinsic reward, and in doing so collects additional transitions that serve as training data for the dynamic models for the subsequent training phase.

For *Disagreement* (Pathak et al., 2019), we follow the author's implementation, changing minimal details. The intrinsic reward is formulated as the predictive variance of the approximate dynamic model, where the model is given by a bootstrap ensemble. In this work, we report results using two versions of this method. The proposed intrinsic reward specifically is formulated in a manner quite similar to our own, however, an ensemble is used instead of a distribution for the approximate posterior. In section §4 we report results *only using the intrinsic reward*, instead of the full proposed method which makes use of a differentiable reward function, which treats the reward as a supervised loss. We do this because the form of the approximate dynamic model does not preclude the use of different policy optimization techniques. Nonetheless, in the next section §A.2 we report results using the full method as proposed, on each continuous control experiment.

The Bayesian approach is extended in (Houthooft et al., 2016)

## A.2 Additional Results

Here we report additional results on both our toy chain task. We also perform additional comparisons with *Disagreement*, including the original policy optimization method with a differentiable reward function as in (Pathak et al., 2019).

### A.2.1 Extended NChain Results

Here we show the results of each method on the NChain task. We used the same experimental setup as in section 4.1 for our experiments. For both *ICM* and *Disagreement* we again use MCTS with UCB-1 to find exploration policies, with the same hyperparameters used for our method and *MAX*. We find that reducing uncertainty is a crucial component to be able to explore the chain. *ICM* only reduces prediction error, so it is not interested in states it can predict correctly. The only source of reward for *ICM* is in the flipped directions of some states, so its possible to initialize chains where *ICM* cannot find novel states, causing high variance between runs. We found that *Disagreement* performs better than *ICM* but not as well as *MAX* or our method. We suspect that this is due to a weakness in the ensemble diversity, lessening the effect of the intrinsic reward for this task. The

---

[4]Swish refers to the nonlinearity proposed by (Ramachandran et al., 2017) which is expressed as a scaled sigmoid function: $y = x + sigmoid(\beta x)$

ensemble can easily overfit to a toy environment like NChain, reducing its ability to explore. *MAX* may avoid overfitting to the chain due to using ensembles of stochastic neural networks. Our method uses amortized SVGD to promote model diversity, and we use an estimate of the uncertainty in our dynamic model to explore new states.

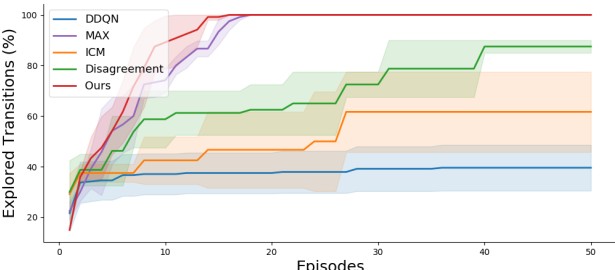

Figure 7: Results on the 40-link chain environment. Each line is the mean of three runs, with the shaded regions corresponding to ±1 standard deviation. Both our method and *MAX* actively reduce uncertainty in the chain, and therefore are able to quickly explore to the end of the chain. $\epsilon$-greedy DDQN fails to explore more than 40% of the chain. Both *ICM* and *Disagreement* perform better but explore less efficiently.

### A.2.2    EXTENDED DISAGREEMENT RESULTS

We repeat our main experiments, comparing our method to both disagreement purely as an intrinsic reward, as well as the full method using the differentiable reward function for policy optimization. For the full method we use the author's official implementation for the following experiments. This is in contrast to the method reported in the main text where we only implement the intrinsic reward. Figures 8, 9, and 10 show results on the Acrobot, Ant Maze, and Block Manipulation environments, respectively. In each figure, lines correspond to the mean of five seeds, and shaded regions denote ± one standard deviation. In each experiment, we can see that treating the intrinsic reward as a supervised loss (gray) improves on the baseline disagreement intrinsic reward (green). However, our method (red) remains the most sample efficient in these experiments.

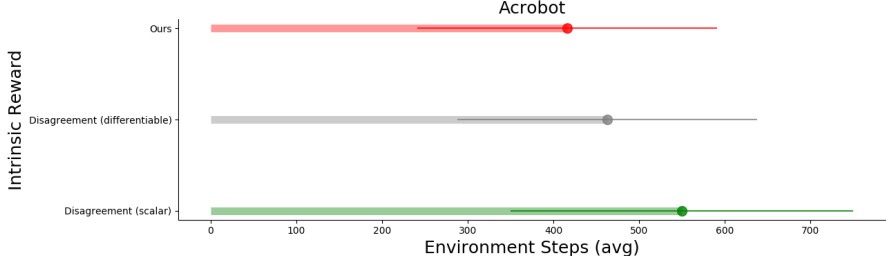

Figure 8: Performance of each exploration method on the Acrobot environment.

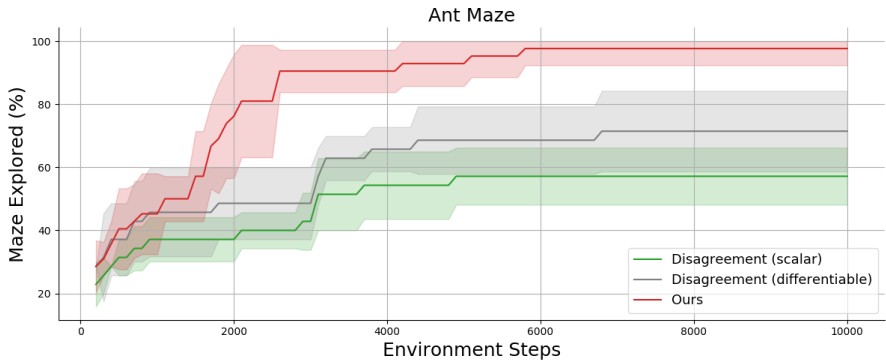

Figure 9: Performance of each exploration method on the Ant Maze environment.

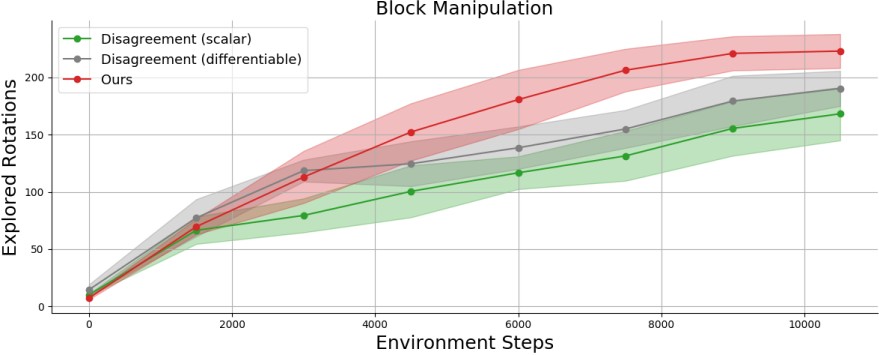

Figure 10: Performance of each exploration method on the Block Manipulation environment.

