# OpenReview forum: "Implicit Generative Modeling for Efficient Exploration"
_ICLR.cc/2020/Conference — Reject_

### Official Review · AnonReviewer1 · 2019-10-21
**Official Blind Review #1**

**Rating:** 3

**Review:**

Update: I thank the authors for their response. I believe the paper has been improved by the additional baselines, number of seeds, clarifications to related work and qualitative analysis of the results. I have increased my score to 3 since I still have some concerns. I strongly believe the baselines should be tuned just as much as the proposed approach on the tasks used for evaluation. The baselines were not evaluated on the same environments in the original papers, so there is not much reason to believe those parameters are optimal for other tasks. Moreover, the current draft still lacks comparisons against stronger exploration methods such as Pseudo-Counts (Ostrovski et al. 2017, Bellemare et al. 2016) or Random Network Distillation (Burda et al 2018).

Summary:
This paper proposes the use of a generative model to estimate a Bayesian uncertainty of the agent’s belief of the environment dynamics. They use draws from the generative model to approximate the posterior of the transition dynamics function. They use the uncertainty in the output of the dynamics model as intrinsic reward.

Main Comments:

I vote for rejecting this paper because I believe the experimental section has some design flaws, the choice of tasks used for evaluation is questionable, relevant baselines are missing, the intrinsic reward formulation requires more motivation, and overall the empirical results are not convincing (at least not for the scope that the paper sets out for in the introduction).

While the authors motivate the use of the proposed intrinsic reward for learning to solve tasks in sparse reward environments, the experiments do not include Moreover, some of the tasks used for evaluation do not have very sparse reward (e.g. acrobot but potentially others too). Without understanding how this intrinsic reward helps to solve certain tasks, it is difficult to assess its effectiveness. While state coverage is important, the end goal is solving tasks and it would be useful to understand how this intrinsic reward affects learning when extrinsic reward is also used. Some types of intrinsic motivation can actually hurt performance when used in combination with extrinsic reward on certain tasks.

I am not sure why the authors chose to not compare against VIME  (https://arxiv.org/pdf/1605.09674.pdf) and NoisyNetworks (https://arxiv.org/pdf/1706.10295.pdf) which are quite powerful exploration methods and also quite strongly related to their our method (e.g. more so than ICM).

Other Questions / Comments:

1. You mention that you use the same hyperparameters for all models. How did you select the HPs to be used? I am concerned this leads to an unfair comparison given that different models may work better for different sets of HPs. A better approach would be to do HP searches for each model and select the best set for each.
2. Using only 3 seeds does not seem to be enough for robust conclusions. Some  of your results are rather close
3. How did you derive equation (1)? Please provide more explanations, at least in the  appendix.
4. Why is Figure 3 missing the other baselines: ICM & Disagreement? Please include for completeness
5. Please include the variance across the seeds in Figure 4 (b).
6. How is the percentage of the explored maze computed for Figure 5? Is that across the entire training or within one episode? What is the learned behavior of the agents? I believe a heatmap with state visitation would be useful to better understand how the learned behaviors differ within an episode. e.g. Within an episode, do the agents learn to go as far as possible from the initial location and then explore that “less explored” area or do they quasi-uniformly visit the states they’ve already seen during previous episodes?
7. In Figure 6 (b), there doesn’t seem to  be a significant difference between your model and the MAX one. What happens if you train them for longer, does MAX achieve the same or even more exploration performance as  your model? I’m concerned this small difference may be due to poor tuning of HPs for the baselines rather than algorithmic differences?
8. For the robotic hand experiments, can you  provide some intuition about what the number of explored rotations means and how they relate to a good policy? What is the number of rotations needed to solve certain tasks? What kinds of rotations do they explore -- are some of them more useful than others for manipulating certain objects? This would add context and help readers understand what those numbers mean in practice in terms of behavior and relevance to learning good / optimal policies.


**Experience Assessment:**

I have published one or two papers in this area.

**Review Assessment: Checking Correctness Of Derivations And Theory:**

I assessed the sensibility of the derivations and theory.

**Review Assessment: Checking Correctness Of Experiments:**

I carefully checked the experiments.

**Review Assessment: Thoroughness In Paper Reading:**

I read the paper thoroughly.

---

> ### Author Response · Authors · 2019-11-12
> **Thank you for your comments**
>
> Thank you for taking the time to review our paper. Below, we answer each question asked by the reviewer, and we will add further discussion/content to the paper.
>
> Q1: Why aren’t external rewards considered?
>
> We study the exploration in the task-agnostic exploration setting, from this perspective, all unseen states are equally valuable to explore, since they all may correspond to a goal state for a potential task. For future work, we will focus this exploration using extrinsic rewards, but it is important to know first if the intrinsic reward is enough to cause the agent to adapt to environmental difficulties.
>
> Q2: Why not compare against NoisyNets [4]?
>
> [4] is functionally very different from our method; adding noise to the model parameters doesn't preclude the use of other exploration methods. As we explore model-based approaches to efficient exploration, we believe comparing with more similar approaches is more informative. We could always add NoisyNets to any of the methods in our paper.
>
> Q3: Why not compare against VIME [5]?
>
> We believe VIME is a principled method that's worth comparing against. VIME uses BNNs to maintain a probabilistic model of the environment dynamics. The predictions of the model are used to estimate compression improvement (a form of Bayesian information gain), and therefore the novelty of states. We believe that this paradigm and method are well represented in [2], which we compare against. Furthermore, [2] is more closely related to our method (by using an ensemble of probabilistic models), and thus the comparison is more informative.
>
>
> 1) How were the hyperparameters chosen?
>
> We used the hyperparameters from the code provided by the authors of [2]. As such, the choice of hyperparameters is not biased towards our method.
>
>
> 2) Random Seeds.
>
> We will run more trials with different seeds and add them to the paper.
>
>
> 3) Equation 1.
>
> In equation 1 we state the variance of predictions given by an ensemble of models. We use this variance to be a measure of the model’s uncertainty
>
>
> 4) Other baselines for chain task.
>
> We will run [2] and [3] on the chain task and update the paper.
>
>
> 5) Error bars on Acrobot experiment.
>
> We will include error bars on the Acrobot experiment and update the paper.
>
>
> 6) Ant Maze: how is the percentage of the maze explored calculated?
>
> We follow [2] and present the percentage of the maze explored during the entire run. We will add a plot showing state visitation behavior by episode for the maze.
>
>
> 7) Closeness to [2] in the robotic manipulation task.
>
> As stated above, the hyperparameter selection was taken from the code provided by the authors, likely putting our method at a disadvantage instead. With regard to long-horizon behavior vs [2], the experiment is testing for efficient exploration in a difficult environment, not for time-to-complete. In difficult environments where the reward is not present, the most relevant task for the agent is to quickly explore the environment. Our experiments focus on this and show that our method is effective.
>
> 8) How do we interpret the agent's performance on the robotic hand experiments?
>
> We characterize a state as a rotation of the held block, without considering the positions of the joints. We discretize the possible block rotations into 512 possible states for this task, meaning that each state is a 45-degree increment in the x, y, z directions. The best agent (ours) explores approximately half the space. This is due to the much higher state/action dimensionality of this environment, and to a greater degree than Ant Maze, states are not uniformly accessible.
>
> In the task agnostic setting, we do not consider the effect on downstream policies. Such a question is left for future work. We examine how our intrinsic reward motivates agents to overcome environmental difficulties, such as the need to learn a skill before encountering novel states.
>
> [1] Gangwani, Tanmay, Qiang Liu, and Jian Peng. "Learning Self-Imitating Diverse Policies". International conference on learning representations. 2019.
> [2] Shyam, Pranav, Wojciech Jaśkowski, and Faustino Gomez. "Model-Based Active Exploration." International conference on machine learning. 2019.
> [3] Pathak, Deepak, Dhiraj Gandhi, and Abhinav Gupta. "Self-Supervised Exploration via Disagreement." International conference on machine learning. 2019.
> [4] Eysenbach, Benjamin, et al. "Diversity is all you need: Learning skills without a reward function." arXiv preprint arXiv:1802.06070 (2018).
> [5] Houthooft, Rein, et al. "Vime: Variational information maximizing exploration." Advances in Neural Information Processing Systems. 2016.

---

### Official Review · AnonReviewer2 · 2019-10-24
**Official Blind Review #2**

**Rating:** 3

**Review:**

Update: I thank the authors for their rebuttal. Having read the other reviews I still stand by my assessment and agree with the other reviewers that the empirical validation should stronger, adding more baselines and conducting experiments on the same environments as your main competitors for fair comparison.

Summary
This paper proposes a Bayesian approach for modeling the agent's uncertainty about forward predictions in the environment, that is, given a state and action how likely the next state is. The uncertainty is then used to define an intrinsic reward function. The paper has sufficient technical depth. However, I am disappointed by the comparison to prior work.

Strengths
Interesting non-parametric approach to estimating uncertainty in the agent's forward dynamics model
Clearly written paper with sufficient technical depth
Well structured discussion of related work

Weaknesses
My main problem with the paper is a missing fair comparison to prior work. The two main contenders are MAX by Shyam et al 2019 and the Disagreement approach by Pathak et al 2019. Comparing the results on AntMaze presented here with those in Shyam I see that MAX only gets to high 80s in terms of maze exploration rate, while in their paper it is in the 90s. In comparison to Pathak, as far as I understand, a different robotic manipulation task was used (HandManipulateBlock here in comparison to the grasp and push objects on a table task by Pathak). Moreover, there are no experiments comparing the proposed approach to the stochastic Atari environments investigated in Pathak et al 2019. I understand this would require dealing with discrete action spaces, but I don't see why this would be infeasible. Overall, I believe this makes it hard to draw conclusions with respect to Shyam et al and Pathak et al and adding these missing comparisons would strengthen the paper substantially in my view.

Minor
p3: "Let f denote the dynamics model" – I believe it would be good to mention the signature of this function (it can be inferred from Figure 1, but it would be nice to make this explicit).
Questions to Authors

**Experience Assessment:**

I do not know much about this area.

**Review Assessment: Checking Correctness Of Derivations And Theory:**

I assessed the sensibility of the derivations and theory.

**Review Assessment: Checking Correctness Of Experiments:**

I assessed the sensibility of the experiments.

**Review Assessment: Thoroughness In Paper Reading:**

I read the paper at least twice and used my best judgement in assessing the paper.

---

> ### Author Response · Authors · 2019-11-12
> **Thank you for the comments**
>
> Thank you for taking the time to review our paper. We clarify some points below, and we will add further discussion to the paper.
>
> Q: The paper shows different results than [1] on the Ant Maze environment.
>
> We used the GitHub repository provided by the authors to run our experiments and kept all the hyperparameters the same. Our reported mean is within the standard deviation reported in the MAX paper, if only just. We agree that more random seeds should be run for all methods, and we will add them to the paper. This should clear up any discrepancies.
>
> Q: The robotic manipulation task is different from the one in [2].
>
> We believe our task encompasses the strengths of the FetchPush environment used in [2] for a much lower computational cost. In the FetchPush experiment, the agent is evaluated based on its interaction rate with an object on the table. Presumably, the point of the task is that a skill must be learned before further exploration can take place. In the same way, we can also infer that an agent which explores the state-space of the HandManipulateBlock environment has learned to accurately manipulate the held block.
>
> Q: There is no comparison to the stochastic Atari environments used in [2].
>
> We think that the experiments proposed in [1] are sensible (chain and ant) with regard to assessing efficient exploration i.e. in less than hundreds of millions of steps, and so we continued this line of work. Because we studied efficient exploration, we avoided large scale experiments like Atari, in favor of simpler, difficult tasks like navigation and manipulation. In contrast to Atari environments, performance on our tasks adds to intuitions about the agent’s behavior with our model.
>
> [1] Shyam, Pranav, Wojciech Jaśkowski, and Faustino Gomez. "Model-based active exploration." International conference on machine learning. 2019.
> [2] Pathak, Deepak, Dhiraj Gandhi, and Abhinav Gupta. "Self-Supervised Exploration via Disagreement." International conference on machine learning. 2019.

---

### Official Review · AnonReviewer3 · 2019-10-25
**Official Blind Review #3**

**Rating:** 3

**Review:**

Summary:

This paper introduces a new intrinsic reward for aiding exploration. This one
is based on learning a distribution on parameters for a neural network which
represents the dynamic function. The variance the predictions from this
dynamic function serves as the intrinsic reward. Results are compared against
several current state-of-the-art approaches.

Feedback:

Using uncertainty as an instrinc reward to guide exploration is a
very active area of research and it would have been more helpful
to say how this work differs from Burda et al, Eysenbach et al,
Gregor et al. 1.  The underlying algorithms are all very similar
and differ in only small and subtle ways. The main difference
with this work and Pathak et al. seems to be that the variance is
all coming from one particular conditional distribution rather than
an ensemble of models, but in Pathak et al it is also a distribution
over models.

Amortized SVGD is used instead of regular SVI in this work, but
it is never articulated why to problem benefits from using that
framework. This paper would greatly benefit from some explanation.
It is mentioned as a novel aspect of the work, but never really
justified at all.

The experimental results and convincing and do show a substantial
improvements over similar approaches in domains in ant maze
navigation and robot hand.

[1] Gregor, Karol, Danilo Jimenez Rezende, and Daan
Wierstra. "Variational intrinsic control." arXiv preprint
arXiv:1611.07507 (2016).


**Experience Assessment:**

I have read many papers in this area.

**Review Assessment: Checking Correctness Of Derivations And Theory:**

N/A

**Review Assessment: Checking Correctness Of Experiments:**

I carefully checked the experiments.

**Review Assessment: Thoroughness In Paper Reading:**

N/A

---

> ### Author Response · Authors · 2019-11-12
> **Thank you for the comments**
>
> Thank you for taking the time to review our paper. We clarify some points below, and we will add further discussion to the paper.
>
> Q: How does this work differ from the following papers on intrinsic rewards?
>
> * Burda et al. (we assume the reviewer means [1])
> Burda et al. [1] is an extension method to ICM, which we directly compare against in our paper (section 4.2). The difference between [1] and ICM is the choice of feature space in which to compute the prediction error.
>
> Our method uses the variance in predictions given by samples from an implicit ensemble, as opposed to using the error of a single model. Our intrinsic reward minimizes the model's uncertainty of the transition function, where prediction error has no notion of model uncertainty.
>
>
> * Eysenbach et al. (DIAYN) [2] and Gregor et al (VIC) [3]
> [2] directly optimizes policy diversity using a maximum entropy policy for the purpose of learning discrete skills. Skill diversity is further enforced by using a discriminator to infer the specific skill from the states visited by the agent. This is somewhat similar to [3], which presents a variational lower bound to the empowerment criterion.
>
> In contrast, we directly minimize the model's uncertainty of the transition function. This encourages the model to directly explore the state space where the model predictions reflect "internal" conflict.
>
> Q: What is the difference between this method and [4]?
>
> * We would like to clarify the main novelty in this paper is in training for a distribution of models, optimized with amortized SVGD, instead of an explicit ensemble as in [4]. We believe that our implicit model can offer more flexible approximations to the true posterior of the dynamic model than the explicit ensemble in [4], because:
> With amortized SVGD, we have inherent model diversity during training. As opposed to using bootstrapping as in [4,5] to avoid collapsing to the MAP solution, which would result in no diversity. Bootstrapping also may fail to cover the entire support of the model posterior.
>
> * We only need to train one model posterior, and then can sample any number of diverse models from it, while [4] needs to train each of its ensemble models separately.
> The benefit of using an implicit distribution over models can be observed from our ability to explore faster across all tasks.
>
> Q: What are the benefits of using the Amortized SVGD framework?
>
> As we state in section 1, using Amortized SVGD allows us to learn a more flexible approximate posterior than would be possible with (stochastic) VI. Particle-based variational inference allows us to avoid assuming a certain parametric form of the posterior (so that the KL divergence is known analytically) or performing MCMC. With SVGD we can compute the true gradient of the KL divergence between our samples and the posterior, instead of just optimizing a lower bound (ELBO) [6]. Further, amortizing SVGD by using a generator increases flexibility since we are not tied to using a fixed number of particles, but can sample any number of models directly.
>
> [1] Burda, Yuri, et al. "Large-Scale Study of Curiosity-Driven Learning". International conference on learning representations. 2019.
> [2] Eysenbach, Benjamin, et al. "Diversity is all you need: Learning skills without a reward function." International conference on learning representations. 2019.
> [3] Gregor, Karol, Danilo Jimenez Rezende, and Daan Wierstra. "Variational intrinsic control." International conference on learning representations. 2017.
> [4] Pathak, Deepak, Dhiraj Gandhi, and Abhinav Gupta. "Self-Supervised Exploration via Disagreement." International conference on machine learning. 2019.
> [5] Shyam, Pranav, Wojciech Jaśkowski, and Faustino Gomez. "Model-based active exploration." International conference on machine learning. 2019.
> [6] Liu, Qiang, and Dilin Wang. "Stein variational gradient descent: A general purpose bayesian inference algorithm." Advances in neural information processing systems. 2016.

---

### Author Response · Authors · 2019-11-15
**Summary of revisions to the paper**

We would like to thank the reviewers for the helpful comments and suggestions. We have replied to each of the reviewer's comments, and now we have incorporated much of the suggested revisions, including both additional discussion and experimental results. The revisions to the paper are as follows:

`1) Added results with 5 random seeds to all methods and experiments. We believe these results are faithful to the methods we compare against.
2) Added state visitation charts to the Ant Maze experiment in section 4.2.2, figure 5.
    * The figures show how the agent moves through the maze throughout the duration of each episode.
    * Multiple figures (c-f) show how this develops with more training.
3) We added results from (Pathak et al. 2017) and (Pathak et al. 2019) on the toy NChain task to Appendix A.2.1.
4) Improved explanation of robot manipulation task, explaining why the task is difficult and work examining.
5) Added or expanded the discussion of some related works, notably (Gregor et al. 2016), (Houthooft et al. 2016), and (Fortunato et al. 2017).
6) Finally, we incorporated the minor changes suggested by the reviewers.

---

### Decision · Program_Chairs · 2019-12-19

**Decision:**

Reject

**Comment:**

There is insufficient support to recommend accepting this paper.  The authors provided detailed responses, but the reviewers unanimously kept their recommendation as reject.  The novelty and significance of the main contribution was not made sufficiently clear, given the context of related work.  Critically, the experimental evaluation was not considered to be convincing, lacking detailed explanation and justification, and a sufficiently thorough comparison to strong baselines, The submitted reviews should help the authors improve their paper.